

# The relationship between functional movement patterns, dynamic balance and ice speed and agility in young elite male ice hockey players

Małgorzata Grabara[1] and Anna Bieniec[2]

[1] Institute of Sport Science, the Jerzy Kukuczka Academy of Physical Education, Katowice, Poland
[2] Department of Health-related Physical Activity and Tourism, the Jerzy Kukuczka Academy of Physical Education, Katowice, Poland

## ABSTRACT

**Background:** Understanding the relationship between the functional state of the musculoskeletal system and skating performance in ice hockey players is essential, as it can provide valuable insights for the development of training programs tailored to the specific needs of athletes. This study investigated the relationship between functional movement patterns, dynamic balance, and ice speed and agility in young elite male ice hockey players.

**Methods:** The study involved sixty elite male ice hockey players aged 14 to 18 years, with an average age of $15.9 \pm 0.85$ years and training experience ranging from 7 to 9 years. Functional movement patterns were evaluated using the Functional Movement Screen™ (FMS™). Dynamic balance was assessed using the lower quarter Y-Balance test (YBT-LQ). Fitness tests on ice were conducted using a professional Smart Speed measurement system.

**Results:** Negative correlations were found between the in-line lunge and the results of the 5-m forward (rho = $-0.31$, $p = 0.018$) and 5-m backward (rho = $-0.27$, $p = 0.040$), as well as between the hurdle step and the 30-m forward skating test result (rho = $-0.26$, $p = 0.043$). Positive correlations were observed between shoulder mobility and both forward (5-m: rho = 0.27, $p = 0.035$) and backward skating results (5-m: rho = 0.35, $p = 0.006$; 30-m: rho = 0.26, $p = 0.047$), and between active straight leg rise and both the 5-m forward skating (rho = 0.38, $p = 0.002$) and agility tests (rho = 0.39, $p = 0.002$). The study also revealed positive correlations between the magnitude of asymmetries in the anterior reach distance of the right and left legs and the results of 5-m forward (rho = 0.34, $p = 0.009$) and backward skating (rho = 0.32, $p = 0.013$). Additionally, a positive correlation was found between the agility test and the magnitude of asymmetries in the posteromedial reach distance (r = 0.32, $p = 0.012$) as well as the composite YBT score (r = 0.28, $p = 0.031$). Negative correlations were found between normalized reach distances in the YBT-LQ and performance outcomes in both forward and backward skating, as well as in the agility test, indicating that greater reach distance corresponds to faster skating.

**Conclusions:** These findings suggest the potential impact of balance and hip mobility on skating speed and agility and emphasize the importance of symmetry for optimal performance among ice hockey players.

Corresponding author
Małgorzata Grabara,
m.grabara@awf.katowice.pl

# INTRODUCTION

Ice hockey, being the fastest sport in professional athletics, is a challenging game. It's a high-intensity team sport that requires players to exhibit rapid skating, swift direction changes, abrupt braking, and engage in substantial body contact (*Montgomery, 2006*). Players typically have a shift lasting approximately 1 min, followed by a retreat to the box for recuperation. During each shift, players are expected to exert maximum effort and speed. Acceleration and sprint speed abilities are crucial skills influencing hockey players' performance (*Montgomery, 2006*; *Stastny et al., 2023*).

The sport of ice hockey requires the development of muscular strength, anaerobic power, speed, and agility (*Montgomery, 2006*). The motor foundation, encompassing coordination, mobility, and stability, forms the basis for both the specific movement patterns essential for strength and speed training, as well as the sport-specific patterns required in a particular sport (*Boyle, 2016*).

Ice hockey players should be assessed using both on-ice and off-ice testing variables. On-ice skating tests reflect mode-specific adaptations, while off-ice tests enable strength and conditioning professionals to design training programs that support on-ice skating performance (*Boland, Delude & Miele, 2019*). Additionally, testing ranges of motion and muscle force balance between agonists and antagonists can assist in training planning, promote more ergonomic skating, and help prevent overload and injury (*Upjohn et al., 2008*). The Functional Movement Screen[TM] (FMS[TM]) and lower quarter Y-Balance (YBT-LQ) tests are frequently employed as rapid screening tools for assessing athletes' functional performance, predicting injury risk, and evaluating the effectiveness of training programs cross various sports disciplines (*Song et al., 2014*; *Plisky et al., 2009*). The FMS[TM] is used to evaluate neuromuscular control, stability of the trunk, and asymmetry between sides (*Chalmers et al., 2017*; *Sikora & Linek, 2022*). The results can be used to identify deficiencies in fundamental movement patterns and to detect left-right asymmetries that occur during these movements (*Cook et al., 2014a*; *Rowan et al., 2015*). On the other hand, the YBT has been used to measure dynamic balance and side-to-side asymmetry (*Plisky et al., 2009*). A study by *Sikora & Linek (2022)* identified weak to moderate correlations between total FMS[TM] and YBT-LQ scores. However, the authors cautioned against using these tests interchangeably due to their assessment of distinct movement deficits.

Most studies employing the FMS[TM] protocol, without the inclusion of a YBT-LQ evaluation, have been conducted on physically fit individuals. This primarily encompasses athletes engaged in individual sports and team-based games, in addition to military personnel and firefighters (*Kiesel, Plisky & Voight, 2007*; *Kiesel, Plisky & Butler, 2009*; *Schneiders et al., 2011*; *Frost et al., 2012*; *Saki, 2017*; *Goss et al., 2009*). Previous studies have demonstrated that differences in functional movement patterns and balance ability are evident across various sports and within different performance levels of a specific sport (*Butler et al., 2012*; *Rokaya et al., 2021*; *Kramer et al., 2019*; *Lisman et al., 2018*).
To date, relatively few studies have been published that utilize both FMS$^{TM}$ and Y-Balance tests among hockey players (*Rowan et al., 2015*; *Boguszewski et al., 2017*). In the study by *Bieniec & Grabara (2024)*, the FMS$^{TM}$ and YBT-LQ were used to determine the effect of a 12-week functional training program on functional movement patterns and dynamic balance in young elite ice hockey players. *Rowan et al. (2015)* focused on the identification of correlations between FMS$^{TM}$ scores and other medical, physical, and physiological fitness assessment outcomes among elite junior ice hockey players. Other studies have analyzed the functional state of the musculoskeletal system of hockey players in the context of injury rates, assessed core stability, and determined age-related differences in the unilaterality of limb movements in hockey players using FMS and YBT-LQ tests (*Kokinda et al., 2020*, *2018*). In another study, the authors evaluated functional movement patterns and spinal posture of elite ice hockey players and examined the association between spinal posture, prevalence of musculoskeletal symptoms and FMS$^{TM}$ scores (*Grabara & Bieniec, 2023*).

A few studies have investigated the association between FMS$^{TM}$, YBT-LQ and physical performance. *Francavilla et al. (2021)* examined the effects of strength and agility training on FMS$^{TM}$ scores in basketball players and investigated whether improvements in FMS$^{TM}$ scores were associated with enhancements in performance. *Blanár et al. (2020)* explored the relationship between the skating and running performance of a youth category ice hockey player, the explosive strength of their lower limbs, and dynamic balance using YBT-LQ. The available evidence has not definitively established whether FMS$^{TM}$ scores truly correlate with or predict athletic performance. Previous studies have suggested that FMS$^{TM}$ is weakly correlated with athletic performance (*Francavilla et al., 2021*; *Okada, Huxel & Nesser, 2011*).

Despite the existence of numerous studies, there is a dearth of research exploring dependencies between the assessment of the functional state of the musculoskeletal system and skating performance among ice hockey players. Furthermore, the literature reveals weak or non-existent associations between the total FMS$^{TM}$ score and athletic performance, with a lack of consensus among various studies on this topic. To address this research gap, the objective of the current study was to investigate the relationship between functional movement patterns, dynamic balance, and ice speed and agility in young elite male ice hockey players. We hypothesized that better functional movement patterns and dynamic balance could be related to enhanced skating performance.

## MATERIALS AND METHODS
### Study design
This is a cross-sectional study aimed at investigating the relationship between functional movement patterns, dynamic balance, and on-ice performance in young elite male ice hockey players. The study does not include a control group. Elite level players include those who play for the national under-20 team and are selected for senior ice hockey league games. These elite hockey players participate in national team training camps and may be called up for games with senior teams.

The study was approved by the Bioethics Committee of the Jerzy Kukuczka Academy of Physical Education in Katowice (certificate of approval No. KB 2/2017) and conformed to the standards set by the Declaration of Helsinki. All participants and their parents/legal guardians were informed about the type and the aim of the study, and they provided written informed consent before participating in the study.

## Participants

A total of 65 young male ice hockey players, aged from 14 to 18 years, were screened based on the eligibility criteria. Five participants withdrew from the study due to illness or injury. Consequently, the study included sixty elite male ice hockey players, with a mean age of 15.9 ± 0.85 years, a mean body mass of 80.1 ± 5.69 kg, and a mean body height of 182.2 ± 6.01 cm. All participants were students in the first and second grades at the Sports School of the Polish Ice Hockey Federation. The athletes' training experience ranged from 7 to 9 years.

The criteria for participant inclusion in the study were as follows: a voluntary consent to participate in the study, an absence of injuries sustained within the past 6 months, no absences from competitions exceeding 30 days, and no surgical procedures undergone within the past 12 months. The hockey players had never undergone testing with the FMS[TM] nor the YBT-LQ, and they were not acquainted with these tests.

## Methods and procedures

Anthropometric evaluations were conducted to determine the body height (BH) and body mass (BM) of the participants. An anthropometer was utilized to measure BH, while a Tanita-410 body composition analyzer (Tokyo, Japan) was employed to assess BM.

Fundamental movement patterns were evaluated using the FMS[TM]. As previously described by *Bieniec & Grabara (2024)*, the sub-tests of the FMS[TM] are divided into three global (deep squat, hurdle step, in-line lunge), two mobility (shoulder mobility, active straight leg raise), and two stability patterns (trunk stability push-up, and rotary stability tests). Sub-tests 2–5, and 7 were performed bilaterally to detect potential asymmetries. If the results differed between the right and left sides, the lower score was taken as the final result. Scores ranged from zero to three, with three being the best possible score. The total score was the sum of all sub-tests, with a maximum of 21 points. Zero was given if pain was reported, one if the movement pattern couldn't be completed, two if the movement was performed with compensations, and three for a correct movement. Sub-tests 4, 6, and 7 have additional clearing screens graded as positive or negative (*Cook et al., 2014a*, *2014b*). A positive result leads to a final score of zero for that test. FMS[TM] has shown high intrarater and interrater reliability (*Smith et al., 2013*) and is a reliable test for young elite hockey players (*Parenteau-Goudreault et al., 2014*). The assessment was conducted by a certified physiotherapist with 7 years of experience.

Dynamic balance was assessed by the lower quarter Y-Balance test (YBT-LQ). The participant performed three successful reaches on a Y-shaped device, and the maximal reach distance in each direction was analyzed. If the athlete could not perform three attempts, another attempt was made after 20 s. Reach distance was measured from the

most distal part of the toes to the most distal part of the foot reaching in the anterior, posterior-lateral, and posterior-medial directions. The largest reach scores for each limb in all directions were also included in the analysis. The distances were normalized to the relative length of the lower limb, which was measured from the anterior superior iliac spine (ASIS) to the medial ankle in the supine position using a tape measure. The composite score from three directions of motion was also calculated for the right limb (composite R) and similarly for the left limb (composite L) taking into account the length of the respective limb: composite R = (anterior R + posterolateral R + posteromedial R)/3* the length of right limb/100. Values were expressed in centimeters, and the differences between the right and left leg were provided as absolute values (*Plisky et al., 2009*; *Kokinda et al., 2018*). A difference of 4 cm or more between the reaches of the left and right lower limbs is considered a value above which the risk of injury increases (*Nelson, Wilson & Becker, 2021*). YBT has good to excellent intrarater and interrater reliability and is a reliable test for measuring dynamic balance (*Plisky et al., 2009*).

Fitness tests on ice were conducted using a professional Smart Speed measurement system (Fusion Sport, Coopers Plains, QLD, Australia). The system includes gates with photocells and a reader for athlete identification. A participant can start at any time from a standing position and the time is counted from the moment he passes the first photocell. Both forward and backward tests were performed twice, with the best result used for analysis. Four tests from the International Ice Hockey Federation (IIHF) database were used to assess specific on-ice fitness elements. These included 30-m straight-line speed tests on ice (forward and backward skating) and agility tests (*Wagner et al., 2021*; *Daigle et al., 2022*). The speed tests used four gates placed at the starting line, and 5, and 15 m apart. The final four gates, located 30 m from the start, marked the finish line. The two intermediate times (5, 15 m) and the total final time (30 m) were considered. The agility test involved a complex sequence of forward and backward skating, with the start line doubling as the finish line. Only one photocell was used for timing, placed 2 m away from the light reflector. The athlete also completed a full lap of the rink behind the net lines as quickly as possible (Full speed test). The test used one photocell set at the start/finish line.

## Statistical analysis

The required sample size for this investigation was determined using G*Power software (version 3.1.9.7), developed by Heinrich Heine University in Düsseldorf, Germany. For correlation analyses, the computation parameters included an anticipated effect size of 0.36, an alpha error probability of 0.05, and a test power of 0.9. For logistic regression, the parameters included an anticipated odds ratio of 1.9, an alpha error probability of 0.05, and a test power of 0.9. These specifications resulted in a required total sample size of fifty-nine.

The results were expressed as means (M) with standard deviations (SD), medians, confidence intervals (CI −95% to 95%), minimum and maximum values (min-max), or described using means with standard deviations, medians, lower and upper quartiles, and frequencies (n). The normality of distributions was tested using the Kolmogorov-Smirnov

test, and the resulting *p*-values are presented alongside the results in the corresponding tables. A *p*-value less than 0.05 indicated a lack of normality of the distribution.

Associations between FMS$^{TM}$ results (sub-tests and total FMS$^{TM}$ score) and skating performance (ice speed and agility tests) were assessed using the Spearman's correlation coefficient. Associations between YBT-LQ results and ice speed and agility tests were assessed using Pearson's or Spearman's correlation coefficients. The Pearson's correlation coefficient (r) and the Spearman's rank correlation (rho) were qualitatively evaluated as follows: below 0.2–very weak correlation (practically no relationship), 0.2 to < 0.4 as weak correlation, 0.4 to < 0.6 as moderate, 0.6 to < 0.8 as strong, and 0.8 to 1.0 as very strong correlation. Associations between the number of asymmetries and skating performance were assessed using one-way ANOVA or Kruskal-Wallis ANOVA.

A binary logistic regression analysis was conducted, converting the dependent variable, which consisted of the ice speed test results, into a binary variable based on the median. A value of zero indicated a shorter time to complete the test, while a value of one indicated a longer time.

The level of significance in all tests was set at $\alpha = 0.05$. The statistical analysis was performed using Microsoft Excel (Microsoft, Redmond, WA, USA) and STATISTICA$^{TM}$ 13.3 software (TIBCO Software, Palo Alto, CA, USA). The binary logistic regression analysis was conducted using Python, specifically utilizing of statsmodels version 0.13.5 library for statistical modeling.

## RESULTS

The assessment of the functional status of young hockey players is presented in Tables 1 and 2. A total FMS$^{TM}$ score lower than 14 was obtained by 26 (43%) of participants. The most asymmetries (43%) were noted in the in-line lunge test, followed by the shoulder mobility test (32%) and the hurdle step test (30%). The FMS$^{TM}$ sub-tests in which participants achieved the lowest scores were those that test rotatory stability, in-line lunge, and hurdle step. The FMS$^{TM}$ movements for which the hockey players most frequently received the highest score of three were shoulder mobility (37%) and trunk stability push-up (28%).

The greatest reach differences between right (R) and left (L) legs were found in the posterolateral direction. On average, the differences between the right and left sides combined in all directions, considering the length of the lower extremities (composite), and in anterior direction were less than 4 cm. However, the mean differences between the right and left sides in posterolateral and posteromedial directions were more than 5 cm. Asymmetries exceeding 4 cm were observed in 72% of athletes in the lateral direction, in 57% of athletes in the medial direction, and in 43% of athletes in the anterior direction.

The results of the ice speed and agility tests are presented in Table 3.

The analysis of correlations between the FMS$^{TM}$ and ice speed and agility tests revealed some interesting relationships. The results of the 5-m forward skating and the 5-m backward skating were negatively correlated with the in-line lunge (rho = −0.31, *p* = 0.018 and rho = −0.27, *p* = 0.040, respectively). The result of the 30-m forward skating was negatively correlated with the hurdle step (rho = −0.26, p = 0.043). However, we also found

**Table 1 Results of functional movement screening (FMS$^{TM}$) in the participants ($n$ = 60).**

| FMS$^{TM}$ results | Mean (SD) | Median | Lower–upper quartiles | Min–max | 1 point (n) | 2 points (n) | 3 points (n) |
|---|---|---|---|---|---|---|---|
| Deep squat | 1.97 (0.55) | 2 | 2–2 | 1–3 | 10 | 42 | 8 |
| Hurdle step | 1.90 (0.66) | 2 | 1–2 | 1–3 | 16 | 34 | 10 |
| In-line lunge | 1.87 (0.57) | 2 | 2–2 | 1–3 | 14 | 40 | 6 |
| Active straight leg rise | 1.97 (0.55) | 2 | 2–2 | 1–3 | 10 | 42 | 8 |
| Shoulder mobility | 2.30 (0.59) | 2 | 2–3 | 1– 3 | 4 | 34 | 22 |
| Trunk stability push-up | 2.27 (0.48) | 2 | 2–3 | 1–3 | 1 | 42 | 17 |
| Rotary stability | 1.60 (0.49) | 2 | 1–2 | 1–2 | 24 | 36 | 0 |
| Asymmetries | 1.32 (0.98) | 1 | 1–2 | 0–3 | | | |
| FMS$^{TM}$ total score | 13.87 (2.26) | 14 | 12–15 | 9–19 | | | |

**Note:**
   M ± SD, mean and standard deviations.

a positive correlation of shoulder mobility with the results of the 5-m forward skating (rho = 0.27, $p$ = 0.035), 5-m backward skating (rho = 0.35, $p$ = 0.006), and 30-m backward skating (rho = 0.26, $p$ = 0.047), as well as a positive correlation of active straight leg rise with the result of the 5-m forward skating (rho = 0.38, $p$ = 0.002) and agility test (rho = 0.39, $p$ = 0.002). No dependencies were found between skating performance and the number of asymmetries.

The correlation analysis between the YBT-LQ and ice speed and agility tests revealed a positive relationship between the results of the 5-m forward and backward skating, and the magnitude of asymmetries in the anterior reach distance between R and L legs (rho = 0.34, $p$ = 0.009, and rho = 0.32, $p$ = 0.013 respectively), as well as between the result of the 5-m forward skating and the composite score (rho = 0.28, $p$ = 0.029). Additionally, positive correlation was found between the agility test and the magnitude of the posteromedial reach distance between R and L legs (r = 0.32, $p$ = 0.012) as well as the composite YBT score (r = 0.28, $p$ = 0.031). It was also observed that the result of the 5-m forward skating was negatively correlated with normalized scores for the posterolateral R (rho = −0.39, $p$ = 0.002), and L (rho = −0.31, $p$ = 0.017) directions, and the posteromedial R (rho = −0.31, $p$ = 0.017) and L (rho = −0.42, $p$ < 0.001) directions. Similarly, the result of the 5-m backward skating was negatively correlated with the normalized score for the posterolateral R direction (rho = −0.26, $p$ = 0.045) and posteromedial L direction (rho = −0.28, $p$ = 0.031). Furthermore, the results of the 15-m and 30-m backward skating were negatively correlated with the normalized score for the posterolateral L direction (r = −0.26, $p$ = 0.049 and r = −0.29, $p$ = 0.024, respectively). Negative correlations were also found between the agility test and normalized scores for the anterior R (rho = −0.32, $p$ = 0.013), anterior L (rho = −0.35, $p$ = 0.005), posterolateral L (r = −0.40, $p$ = 0.002), and posteromedial L (r = −0.33, $p$ = 0.010) directions. No correlations were observed between the full speed test and the results of reach differences or normalized scores in any direction.

The binary logistic regression analyses with following dependent variables: 5-m forward skating (Table 4), 5-m backward skating (Table 5), and agility (Table 6), showed significant results. Specifically, we observed that a one-unit increase in the asymmetry between

**Table 2 Results of lower quarter Y-balance test (YBT-LQ) in the participants ($n = 60$).**

| YBT-LQ sub-tests | Mean (SD) | Median | CI | Min–max | p-value |
|---|---|---|---|---|---|
| Reach differences between right and left legs (cm) | | | | | |
| Anterior | 3.98 (3.08) | 3 | (3.18–4.77) | 0–12 | 0.02 |
| Posterolateral | 6.12 (4.22) | 6 | (5.03–7.21) | 0–20 | 0.15 |
| Posteromedial | 5.66 (5.42) | 4.5 | (4.26–7.06) | 0–30 | 0.13 |
| Composite | 3.76 (2.76) | 3.32 | (3.05–4.48) | 0.01–10.61 | 0.72 |
| Normalized scores (% leg length) | | | | | |
| Anterior R | 56.98 (9.13) | 56.57 | (54.62–59.33) | 41.41–78.26 | 0.63 |
| Anterior L | 57.31 (10.28) | 58.2 | (54.65–59.96) | 33–83.06 | 0.99 |
| Posterolateral R | 128.57 (14.60) | 127.09 | (124.8–132.34) | 101.98–154.64 | 0.73 |
| Posterolateral L | 128.81 (15.64) | 129.17 | (124.77–132.85) | 94.79–156.25 | 0.90 |
| Posteromedial R | 124.57 (13.69) | 122.92 | (121.03–128.1) | 99.01–175.64 | 0.40 |
| Posteromedial L | 123.98 (15.02) | 124.61 | (120.1–127.86) | 86.87–160 | 0.94 |
| Composite YBT R | 100.04 (9.67) | 98.93 | (97.54–102.54) | 79.46–125.64 | 0.69 |
| Composite YBT L | 99.64 (10.48) | 99.55 | (96.93–102.34) | 79.75–120.49 | 0.95 |

Note:
R, right; L, left; M ± SD, mean and standard deviations; CI, confidence interval (95%); p-value, from Kolmogorov-Smirnov test.

**Table 3 Results of ice speed and agility tests in the participants ($n = 60$).**

| Ice speed tests | Mean (SD) | Median | 95% CI | Min–max | p-value |
|---|---|---|---|---|---|
| 5 m F | 1.35 (0.39) | 1.15 | (1.25–1.46) | (0.76–2.00) | <0.01 |
| 15 m F | 2.81 (0.44) | 2.59 | (2.70–2.93) | (2.23–3.98) | 0.01 |
| 30 m F | 4.30 (0.35) | 4.31 | (4.21–4.39) | (3.18–5.69) | 0.14 |
| 5 m B | 1.78 (0.53) | 1.87 | (1.64–1.92) | (1.03–2.88) | 0.04 |
| 15 m B | 3.23 (0.62) | 3.06 | (3.07–3.39) | (2.25–4.91) | 0.23 |
| 30 m B | 5.15 (0.35) | 5.12 | (5.06–5.24) | (4.31–6.12) | 0.46 |
| Agility | 14.44 (1.00) | 14.57 | (14.18–14.70) | (12.56–18.90) | 0.49 |
| Full speed | 15.30 (0.82) | 15.25 | (15.08–15.51) | (14.01–17.02) | 0.76 |

Note:
F, forward; B, backward; M ± SD, mean and standard deviations; CI, confidence interval (95%); p-value, from Kolmogorov-Smirnov test.

anterior reach of R and L legs increases the log-odds of the 5-m forward skating by 1.62, whereas a one-unit increase in the normalized score in anterior L direction decreases the log-odds of the 5-m forward skating by 2.27 (Table 4). Pseudo-$R^2$ (McFadden) was 0.46, indicating a good fit of the model and explaining 46% of the variance. For the 5-m backward skating, a one-unit increase in the deep squat, in-line lunge, and stability tests decreases the log-odds by 4.85, 4.89, and 6.21 respectively, whereas an increase in total FMS[TM] points increases the log-odds by 3.04 (Table 5). The Pseudo-$R^2$ was 0.51, explaining 51% of the variance. Additionally, a one-unit increase in the asymmetry between the posteromedial R and L directions increases the log-odds of the agility result by 2.34 (Table 6). The Pseudo-$R^2$ was 0.51, indicating a good fit of the model and explaining 51% of the variance.

**Table 4 Logistic regression results with the dependent variable being the time for the 5-m forward skating test.**

| Variable | Coef. | Std. Err. | z | *p*-value | 95% CI lower | 95% CI upper |
|---|---|---|---|---|---|---|
| Constant | −14.95 | 1.54 | −9.7 | 0.00 | −17.97 | −11.93 |
| Asymmetries | −0.17 | 0.53 | −0.31 | 0.757 | −1.21 | 0.88 |
| Reach differences between right and left legs (cm) | | | | | | |
| Anterior | 1.62 | 0.63 | 2.57 | **0.010** | 0.38 | 2.85 |
| Posterolateral | −0.88 | 0.49 | −1.82 | 0.068 | −1.84 | 0.07 |
| Posteromedial | 0.53 | 0.66 | 0.81 | 0.417 | −0.75 | 1.81 |
| Composite YBT | 0.94 | 0.63 | 1.49 | 0.136 | −0.29 | 2.17 |
| Normalized score (% leg length) | | | | | | |
| Anterior L | −2.27 | 0.81 | −2.81 | **0.005** | −3.85 | −0.69 |

Note:
Coef., coefficient; Std. err, standard error; z, z-value from test statistics; *p*-values, less than 0.05 are bolded; CI, confidence interval; L, left.

# DISCUSSION

This study investigated the relationship between functional movement patterns, dynamic balance, and skating performance in young elite male ice hockey players.

The study revealed associations between several FMS™ sub-tests and both ice speed and agility. Specifically, higher scores in the in-line lunge and hurdle step tests, which assess global movement patterns, corresponded to shorter times in both forward and backward skating. Although the observed correlations were generally weak, the regression analysis indicated that better performance in the deep squat, in-line lunge, and stability tests can increase the likelihood of reducing the time required to complete the 5-m backward skating test. These potential associations could offer valuable insights for identifying components of future training programs tailored to the specific needs of athletes. It should also be emphasized that the FMS™ sub-tests associated with better hockey performance were two of the three tests in which the performance of the examined hockey players was poorest. These results are consistent with those obtained by *Okada, Huxel & Nesser (2011)*, who found a negative correlation between the T-Run Agility test and the left in-line lunge among adult male and female recreational athletes. This suggests that a higher agility test score may be associated with better performance in the in-line lunge. *Girard, Quigley & Helfst (2016)* in their review noticed that certain individual components of the FMS™, such as deep squat, did correlate with certain athletic performance measures. *Bennett et al. (2022)* observed a weak negative association between the hurdle step, in-line lunge and agility among adolescent football players. *Lloyd et al. (2015)* found that total FMS™ score, the deep squat, in-line lunge, active straight leg raise, and rotary stability tests were negatively correlated with the ability in soccer players aged 11–16 years. *Koźlenia et al. (2020)* discovered that adult soccer and handball players with lower quality movement patterns (as indicated by the FMS™ score of less than 14) performed worse in agility test. However, the authors did not investigate the relationships between each FMS™ subtest and speed and agility.

**Table 5 Logistic regression results with the dependent variable being the time for the 5-m backward skating test.**

| Variable | Coef. | Std. err. | z | *p*-value | 95% CI lower | 95% CI upper |
|---|---|---|---|---|---|---|
| Constant | −0.39 | 4.67 | −0.08 | 0.933 | −9.54 | 8.75 |
| Deep squat | −4.85 | 2.37 | −2.05 | **0.041** | −9.50 | −0.20 |
| Hurdle step | −1.49 | 1.64 | −0.91 | 0.361 | −4.7 | 1.71 |
| In-line lunge | −4.89 | 1.9 | −2.57 | **0.01** | −8.61 | −1.16 |
| Active leg rise up | −2.06 | 1.52 | −1.36 | 0.175 | −5.04 | 0.92 |
| Stability | −6.21 | 2.77 | −2.24 | **0.025** | −11.64 | −0.77 |
| Rotational stability | −1.47 | 1.41 | −1.04 | 0.298 | −4.23 | 1.3 |
| Total FMS points | 3.04 | 1.38 | 2.21 | **0.027** | 0.34 | 5.74 |
| Asymmetries | 0.04 | 0.68 | 0.06 | 0.954 | −1.3 | 1.376 |
| Reach differences between right and left leg (cm) | | | | | | |
| Anterior | 0.62 | 0.553 | 1.121 | 0.262 | −0.46 | 1.704 |
| Posterolateral | 0.417 | 0.574 | 0.726 | 0.468 | −0.71 | 1.542 |
| Posteromedial | −0.675 | 0.655 | −1.03 | 0.303 | −1.96 | 0.61 |
| Composite YBT | 0.119 | 0.681 | 0.175 | 0.861 | −1.22 | 1.453 |
| Normalized scores (% leg length) | | | | | | |
| Anterior R | 0.46 | 1.117 | 0.41 | 0.681 | −1.73 | 2.65 |
| Anterior L | −0.83 | 1.3 | −0.64 | 0.522 | −3.37 | 1.71 |
| Posterolateral R | −2.44 | 2.49 | −0.98 | 0.328 | −7.32 | 2.45 |
| Posterolateral L | 1.41 | 2.41 | 0.58 | 0.56 | −3.32 | 6.13 |
| Posteromedial R | 1.06 | 1.93 | 0.55 | 0.583 | −2.73 | 4.85 |
| Posteromedial L | −3.54 | 2.5 | −1.42 | 0.156 | −8.44 | 1.35 |
| Composite YBT R | −1.04 | 4.12 | −0.25 | 0.801 | −9.12 | 7.04 |
| Composite YBT L | 3.39 | 4.52 | 0.75 | 0.454 | −5.47 | 12.24 |

**Note:**
Coef., coefficient; Std. err, standard error; z, z-value from test statistics; *p*-values, less than 0.05 are bolded; CI, confidence interval; R, right; L, left.

However, we also identified a few correlations which were seemingly slight, coincidental, or even counterintuitive in their directionality. For example, there was a weak, positive correlation of shoulder mobility with the results of the speed test, and a weak, positive correlation of active straight leg rise with the result of the 5-m forward skating and agility tests. These correlations suggest that a lower score obtained in the mentioned tests is associated with shorter completion times for individual distances in both forward and backward skating, along with enhanced agility. The shoulder mobility test requires mobility in a combination of movements, including flexion/abduction/external rotation, and extension/adduction/internal rotation, along with normal scapular mobility and thoracic spine extension. The active straight-leg raise test necessitates flexibility of the hamstrings, gastrocnemius, and soleus muscles (*Cook et al., 2014b*). Notably, these tests are performed slowly and from positions that are atypical for ice hockey. It is difficult to explain the association between better shoulder mobility and poorer performance on the speed test. However, the relationship between the active straight-leg raise and hockey performance might be explained by the hypothesis that

**Table 6 Logistic regression results with the dependent variable being the agility test.**

| Variable | Coef. | Std. err. | z | p-value | 95% CI lower | 95% CI upper |
| --- | --- | --- | --- | --- | --- | --- |
| Constant | −3.57 | 4.38 | −0.82 | 0.415 | −12.03 | 4.9 |
| Asymmetries | −1.17 | 0.77 | −1.52 | 0.128 | −2.77 | 0.43 |
| Reach differences between right and left legs (cm) | | | | | | |
| Anterior | 0.91 | 0.64 | 1.41 | 0.159 | −0.34 | 2.15 |
| Posterolateral | −0.11 | 0.65 | −0.17 | 0.867 | −1.4 | 1.18 |
| Posteromedial | 2.34 | 1.11 | 2.1 | **0.036** | 0.15 | 4.52 |
| Composite YBT | −1.16 | 1.29 | −0.9 | 0.368 | −2.76 | 0.44 |
| Normalized scores (% leg length) | | | | | | |
| Anterior R | −0.86 | 1.05 | −0.82 | 0.413 | −3.25 | 1.52 |
| Anterior L | 0.73 | 1.14 | 0.64 | 0.524 | −1.52 | 2.97 |
| Posterolateral R | 0.23 | 1.18 | 0.2 | 0.84 | −3.61 | 4.08 |
| Posterolateral L | 0.01 | 1.56 | 0 | 0.997 | −3.88 | 3.89 |
| Posteromedial R | 0.21 | 1.02 | 0.2 | 0.84 | −2.86 | 3.54 |
| Posteromedial L | −0.41 | 1 | −0.41 | 0.678 | −5.11 | 3.33 |
| Composite YBT R | 2.65 | 3.03 | 0.87 | 0.382 | −3.33 | 8.63 |
| Composite YBT L | −1.59 | 3.23 | −0.49 | 0.623 | −7.92 | 4.75 |

Note:
Coef., coefficient; Std. err, standard error; z, z-value from test statistics; p-values, less than 0.05 are bolded; CI, confidence interval; R, right; L, left.

greater flexibility, and by extension greater musculotendinous compliance, may compromise performance in power-based activities such as skating. *Nelson et al. (2005)* observed that stretching exercises performed prior to an activity could potentially have a negative effect on skills that require multiple repetitive high power outputs, in addition to those that primarily depend on maximizing a single output of peak force or power. Our results may also indicate the lack of conformity between the scores on certain FMS[TM] subtests and performance, as revealed in previous studies (*Rowan et al., 2015*; *Okada, Huxel & Nesser, 2011*; *Lockie et al., 2015*). For example, *Okada, Huxel & Nesser (2011)* found a positive correlation between the T-Run Agility test and right shoulder mobility among adult male and female recreational athletes. This correlation may suggest that a shorter time achieved in the agility test is associated with poorer performance in the FMS[TM] sub-test for shoulder mobility. Similarly, *Rowan et al. (2015)* found a link between peak leg power and total FMS[TM] score, which implies that an increase in FMS[TM] total score corresponds to a decrease in low peak leg power. Furthermore, in an assessment of nine female recreational team sport athletes, *Lockie et al. (2015)* observed that those females who performed better on the FMS[TM] also performed worse on the change-of-direction speed tests. Additionally, those females who scored three in the active straight-leg raise tended to be slower in the change-of-direction speed tests.

In conclusion, our study found that a lower quality of the in-line lunge and hurdle step, as well as a higher quality of the shoulder mobility and the active straight-leg raise FMS[TM] sub-tests, was associated with poorer forward and backward skating, and reduced agility. Additionally, a higher total FMS[TM] score decreased the odds of achieving better results in

5-m backward skating test. However, we did not find any relationship between skating performance and a number of asymmetries. *Girard, Quigley & Helfst (2016)* concluded that the total FMS[TM] score is not associated with any aspect of athletic performance, which partially aligns with our results. However, it should be noted that among the studies cited above, only the research conducted by *Rowan et al. (2015)* was carried out among elite junior hockey players. Nevertheless, the authors did not investigate the relationships between FMS[TM] results and ice speed and agility. Therefore, caution should be exercised when interpreting the findings of these comparisons.

The study revealed the relationship between the magnitude of asymmetries in reach distance of right and left legs and the outcomes of forward and backward skating. Specifically, a greater asymmetry in the anterior direction between the right and left legs, as well as in the composite score, was associated with prolonged completion times in the 5-meter forward and backward skating. These asymmetries have a potential to induce compensatory movements in the body, thereby diminishing movement efficiency. Asymmetries in the range of motion (ROM) of the right and left lower limbs may contribute to a reduction in stride length during in skating, which holds particular significance during the initial strides of a sprint and during acceleration. Furthermore, we observed dependencies between normalized reach distances in the YBT-LQ and performance outcomes in both forward and backward skating, as well as in the agility test. Specifically, ice hockey players who demonstrated larger reach distances in the YBT-LQ were able to complete the skating distances more quickly. This highlights the potential impact of balance and hip mobility on skating speed and agility among ice hockey players. Optimal lower limb joint ROM aids in stability by lowering the center of gravity. Increased hip and knee mobility enhances the gluteus maximus stretch-contraction cycle, stimulating concentric quadriceps contraction during push-off, thereby boosting power and acceleration (*Buckeridge et al., 2015*). Similar to these results, *Rokaya et al. (2021)* observed a strong negative correlation between the YBT-LQ composite scores and maximum reach in each direction with the scores of the modified agility T-test among university male soccer players. *Sekulic et al. (2012)* noticed that the balance scores were related to the agility performance for men but not for women, and the authors concluded that the balance should be considered as a potential predictor of agility in adult male athletes. Other studies have posited that balance is a key characteristic of agility, and enhancing balance could potentially improve agility (*Kramer et al., 2019*; *Sporis et al., 2010*). The concept of agility is predicated on an individual's capacity to coordinate and control their center of mass and limbs, thereby facilitating effective acceleration and deceleration during activities such as skating. *Sporis et al. (2010)* noted that by emphasizing agility training, which includes the improvement of balance and coordination, soccer players can enhance their ability to move faster, change directions rapidly, and maintain control.

## Strengths and limitations of this study

The study comprised a relatively large and homogeneous cohort of athletes, encompassing similar age, gender, and the sporting discipline. Our study aimed to identify correlations not only between the overall FMS[TM] score and FMS sub-tests and hockey performance but

also between dynamic balance evaluated by LQ-YBT and hockey performance. However, it is noteworthy that FMS$^{TM}$ scores provide ordinal outcomes, and diverse movement compensations may achieve identical scores in specific assessment, which could affect the strength of any relationships with performance tests (*Lockie et al., 2015*). Furthermore, the FMS$^{TM}$ scoring system has been criticized, with suggestions that it does not provide trainers with specific data about the subject's functionality (*Hernández-García et al., 2020*). Another limitation is the narrow age range of the participants under study; hence, caution should be exercised in generalizing the findings to adult athletes. Additionally, the study did not account for biological age, which should be considered in future research.

## CONCLUSIONS

The study findings indicated that lower quality of the in-line lunge and hurdle step, as well as a higher quality of shoulder mobility and the active straight-leg raise FMS$^{TM}$ sub-tests, are associated with diminished forward and backward skating abilities and decreased agility. Moreover, the study established a correlation between the magnitude of asymmetries in reach distance of the right and left legs and the results of forward and backward skating. Notably, a greater asymmetry in the anterior direction between the right and left legs, as well as in the composite score, correlated with prolonged completion times in the 5-m forward and backward skating. Additionally, dependencies were observed between normalized reach distances in the YBT-LQ and performance outcomes in both forward and backward skating, as well as in the agility test. Specifically, ice hockey players with larger reach distances in the YBT-LQ demonstrated faster completion of skating distances.

Furthermore, the study indicated that greater asymmetry in reach between the right and left sides, and interestingly, higher FMS$^{TM}$ test scores, may decrease the likelihood of faster distance skating performance. Conversely, larger normalized reach distances and better performance in the deep squat, in-line lunge, and stability tests may increase the likelihood of faster distance skating performance.

The study partially supports the hypothesis that better functional movement patterns and dynamic balance are associated with enhanced skating performance. The results highlight the importance of addressing asymmetries and improving balance and mobility for optimal athletic performance in ice hockey.

## ACKNOWLEDGEMENTS

The authors thank the participants and coaches for their invaluable contribution to the study.

### Funding

This study was supported by the publishing fund of Jerzy Kukuczka Academy of Physical Education in Katowice. The funders had no role in study design, data collection and analysis, decision to publish, or preparation of the manuscript.

## Grant Disclosures

The following grant information was disclosed by the authors:
Publishing fund of Jerzy Kukuczka Academy of Physical Education in Katowice.

## Competing Interests

The authors declare that they have no competing interests.

## Author Contributions

- Małgorzata Grabara conceived and designed the experiments, performed the experiments, analyzed the data, prepared figures and/or tables, authored or reviewed drafts of the article, and approved the final draft.
- Anna Bieniec performed the experiments, authored or reviewed drafts of the article, and approved the final draft.

## Human Ethics

The following information was supplied relating to ethical approvals (*i.e.*, approving body and any reference numbers):

Bioethics Committee of the Jerzy Kukuczka Academy of Physical Education in Katowice (certificate of approval No. KB 2/2017).

## Data Availability

The raw data is available in the Supplemental Files.

## Supplemental Information

Supplemental information for this article can be found online at http://dx.doi.org/10.7717/peerj.18092#supplemental-information.

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
