# Peer review of "The relationship between functional movement patterns, dynamic balance and ice speed and agility in young elite male ice hockey players"

_PeerJ, doi:10.7717/peerj.18092_

## Round 0.1 · original submission · Major Revisions

Thank you for submitting your work to PeerJ. Your manuscript has been evaluated by two expert peer reviewers. Both reviewers see value in your study, but list several methodological and statistical issues that should be addressed. I urge you to take a careful look at their comments.

·

Basic reporting

In Additional comments I have added suggestions from this section for authors.

Experimental design

In Additional comments I have added suggestions from this section for authors.

Validity of the findings

In Additional comments I have added suggestions from this section for authors.

Additional comments

Dear Authors,
I write you in regards to manuscript entitled “The relationship between functional movement patterns, dynamic balance and ice speed and agility in young elite male ice hockey players” which you submitted to the PeerJ.
The authors designed a cross-sectional study to investigate the relationship between functional movement patterns, dynamic balance, and ice speed and agility in young elite male ice hockey players. However, the manuscript contains several points that require further attention and may help to improve the research. See below for specific comments.
Check the guidelines of the journal (wording, format and language).
In the “Abstract” section.
− In the ‘Results’ section of the summary show statistical data (e.g. Spearman's rank correlation or p-value).
− In the ‘Background’ section of the abstract show why this research study is important.
− In the Methods section of the abstract use range of data or mean values and standard deviation.
In the “Introduction” section
Please always use two decimal places in all results and tables.
Define the meaning of elite player in ice hockey.
Indicate where the dataset associated with the work is located.
In the “Method” section
Create a Research Design section that includes the type of research design, Bioethics Committee of the Jerzy Kukuczka Academy of Physical Education in Katowice y Declaration of Helsinki.
Lines 61-65. Include this paragraph in the Statistical Analysis section. Calculate the effect size with the study data and perform the a priori calculation. When you have a low sample size usually follow up studies fail to validate original findings.
Lines 66. Delete “In this cross-sectional study,” and detail the sample data in the same sentence (age, body massm height).
Lines 79-83: It does not correspond to this section.
Lines 91-92. Please show data according to dominant and non-dominant body side.
Line 134, 146. Please, use Kolmogorov-Smirnov test for normality because of the sample size of this study. Which independent variables were parametric or non-parametric?
What are the reliability values and experience of the study evaluators?
I suggest performing a predictive or association analysis (cause-effect relationship) e.g. a statistical analysis of univariate and multivariate binary logistic regression.
In the “Results” section
Table 1,2,3,4. Please change M ±SD to Total. Delete the Median column. Please review the journal's guidelines on the layout of Tables and Figures.
The Results section is very long. Focus the text on highlighting the main result of each analysis and Table.
In the “Discusion” section
Please comment on the limitations of the FMS procedure according to the following publication. Hernández-García, R., Gil-López, M. I., Martínez-Pozo, D., Martínez-Romero, M. T., Aparicio-Sarmiento, A., Cejudo, A., ... & Bishop, C. (2020). Validity and reliability of the new basic functional assessment protocol (BFA). International Journal of Environmental Research and Public Health, 17(13), 4845.
In the “Conclusions” section
Please respond directly to the research question and aim.
Thank you for providing the opportunity to review this manuscript.

·

Basic reporting

The article is well written and accessible for readers who are familiar with this area of research. In addition, combining mobility tests and analyzing their associations with performance markers is somewhat interesting and relevant to the field.

The used language (professional English) is adequate, but I think that a linguistic revision might ba an appropriate choice (ex. L8, cultivation; L85, anthropometer; L62, use of parentheses...).

References are appropriate and refer the recent publications. However, I suggest an addition of a paragraph that would present some important information about the relevance of testing ice hockey players in terms of player evaluation, development and processes of strength conditioning approaches that optimizes performance. Many literature exists on this topic, and that would provide a deeper justification.

No major comments in regard with Tables, excepted that they might be reviewed (I would suggest presentation of all. models in a supplementary file info ; why bold characters in Table 5 ?).

I think that the addition of Figures (ex. Y Balance test) would contribute.

Experimental design

Authors achieve good justification on the research questions, by demonstrating a clear knowledge gap in this area of research.

Cross sectional study is the appropriate design. The Methods section is presented in a well-organized fashion, in an acceptable rigorous manner (despite some adjustments).
* * *
However some additional details in regard with Methods need to be clarified:

Experimental design issues:

Line 72: please specify the players' level.
Line 92: lower scores: does this mean the best score of worst (in fact, less assymetry)?
Line 122: 30-m on ice sprint test. OIs it the best distance for testing skating speed ? That can be justified in a deeper way.

L125-130: This short paragraph needs to be clarified to facilitate the study's replicability. An illustration of the skating test might help.

Statistical issues:
L134: Authors present that they intend to verify normality assumptions with SW test. Indeed, no results from this specification is presented. I think they might directly say if normality assumptions were violated. This is the same on Line 156, in which authors specified some asymetry, but evasivelky to me. That might be improved for the reader's benefits.

L137-139: Why performing both Spearman's and / or Pearson's correlations ? If this is for potential violation of normality, this justifies my previous argument (see L134).

L147: variance inflation factor (VIF): Maybe authors should add the range of values that need to be attained for being considered as "satisfying".

Validity of the findings

Results are in line with the study's objectives. Without providing major novelty in terms of results (e.g., they go as expected), the favorable associations between mobility measures and skating performance are important for stakeholders. Mobility assessment certainly can play a role in terms on athletic health, but shedding some light about its associations with performance markers contribute interestingly to support the measurement of mobility.

In terms of the presentation of results, I suggest the addition (instead of simply presenting the only significant models) of all the "non significant" models in supplementary information file - that would allow to see each combination that was analyzed.

Discussion issues:
In general, I found the discussion well articulated and honest (in terms of interpreting the results), covering many relevant points. The raised conclusions are well supported by data from this study. However, I think that there is place for some improvements:

*Potential limitation: I think that physical maturation can play a role in terms of performing in skating test. That was probably not planned in the research design, but taking account for biological maturation (in which many "filed" methods exist) can be an interesting point that might explain performance in a more complete way.

*L248-255: The mechanisms behind the shoulder test mobility and its associations with skating performance is till ambiguous for me (is there really a mechanism?) and authors did not convince me, and maybe there is a need for more clarifications. While this might be outside my specific scope of expertise, I think that some additional insights here might contribute.

Additional comments

I think that my comments in the previous sections cover the more important points.

This is a relevant study that covers an important concept-factor that should be considered in the assessment of players' skating performance. I encourage authors to continue in this line of research, by conducting studies in real settings (game, competition, tournament) in the future.

For being considered for acceptation, I think some areas of the manuscript can be improved which might lead to some important additions to the paper.

---

## Round 0.2 · Minor Revisions

I thank the authors for their thorough work in responding to the concerns of the reviewers. We have solicited additional feedback from Reviewer 2, who only has a few minor suggestions for improved clarity. After the authors respond to these remaining concerns, I would be glad to recommend acceptance.

·

Basic reporting

OVERALL APPRECIATION:
I took some time to revise authors' responses to my previous comments (review 1). In general, I can conclude that they responded in a satisfying manner to me questions and recommendations.

Language - One of my remark or issue that I reported was concerning the quality of language, which was improved in a significant manner.

References in regard with the relevance of hockey testing - I appreciate the fact that authors added a paragraph (or small section) about assessment in ice hockey (see Line 11-15). Since then, the rationale or justification behind the study is well supported.

GENERAL:

Clarifications about tables are now specified. Thank you.

METHODS:

Players' level is now specified.

Line 91: from14 to 18 years... a space should be added between from 14 to 18 years...

Experimental design

Great clarification about the experimental design of the study.

L71-75: *Would it be necessary to specify the country or league level (National Division I, etc...).

L110-120: The explanation about interpreting asymmetry scores is satisfying to me. Authors' response about my previous comment is well justified.

Validity of the findings

L185: Statistical analyses: Authors make an appropriate choice of going towards non parametric model (e.g., logistic regression) because it is in respect with the violation of normality assumptions. From there, it is ok with me to go further with logistic regression instead of linear.

Presenting non-significant models in Supplementary files is appreciated.

DISCUSSION:

L352+: I am still ambivalent about the shoulder mobility associations with results with speed test... maybe shoulder mobility might be an overall valkue for general mobility. In fact, authors brought a realistic explanation, but this not sounds concluding to me.

Additional comments

n/a

---

## Round 0.3 · accepted · Accept

The authors have done an excellent job responding to the remaining concerns of reviewer 2. I especially appreciate the new circumspect language on associations between shoulder mobility and speed in the Discussion section. I am happy to recommend Acceptance.